# Boosting Semi-Supervised Learning via Variational Confidence Calibration and Unlabeled Sample Elimination

## Abstract

Despite the recent progress of Semi-supervised Learning (SSL), we argue that the existing methods may not employ unlabeled examples effectively and efficiently. Many pseudo-label-based methods select unlabeled examples into the training stage based on the inaccurate confidence scores provided by the output layer of the classifier network. Additionally, most prior work typically adpots all the available unlabeled examples without data pruning, which is incapable of learning from massive unlabeled data. To address these issues, this paper proposes two methods called VCC (Variational Confidence Calibration) and INFUSE (INfluence-Function-based Unlabeled Sample Elimination). VCC is a general-purpose plugin of confidence calibration for SSL. By approximating the calibrated confidence through three types of consistency scores, a variational autoencoder is leveraged to reconstruct the confidence score for selecting more accurate pseudo-labels. Based on the influence function, INFUSE is a data pruning method for constructing a core dataset of unlabeled examples. The effectiveness of our methods is demonstrated through experiments on multiple datasets and in various settings. For example, on the CIFAR-100 dataset with 400 labeled examples, VCC reduces the classification error rate of FixMatch from 46.47% to 43.31% (with improvement of 3.16%). On the SVHN dataset with 250 labeled examples, INFUSE achieves 2.61% error rate using only 10% unlabeled data, which is better than RETRIEVE (2.90%) and the baseline with full unlabeled data (3.80%). Putting all the pieces together, the combined VCC-INFUSE plugins can reduce the error rate of FlexMatch from 26.49% to 25.41% on the CIFAR100 dataset (with improvement of 1.08%) while saving nearly half of the original training time (from 223.96 GPU hours to 115.47 GPU hours).

## 1 Introduction

Deep neural networks have become the foundation of many fields in machine learning. The success of neural networks can be partially attributed to the existence of large-scale datasets with annotations such as ImageNet (Deng et al., 2009) and COCO (Lin et al., 2014). However, collecting and labeling a huge amount of data is time-consuming and laborious. Besides, the potential privacy issues are also obstacles to data labeling. In contrast, collecting unlabeled data is cheaper and easier for most tasks.

To mitigate the need for labeled examples, semi-supervised learning (SSL) has been a hot topic in recent years for leveraging cheap and large-volume unlabeled examples. One commonly used approach is **Pseudo-labeling**, which produces artificial pseudo-labels for unlabeled data. For example, FixMatch (Sohn et al., 2020) is one of the most popular such methods.

In FixMatch, the unlabeled data is first fed to the model to get the prediction, followed by a selection module with a fixed threshold to select pseudo labels. Unlabeled data points whose confidence scores are greater than the threshold would be chosen for training, while others are simply ignored. By denoting $\tau$ as the fixed threshold, $c_i$ (or $\widetilde{c}_i$) as the confidence distribution predictions of weakly (or strongly) augmented version of example $i$, respectively, and $\hat{c}_i = \arg\max(c_i)$ as the predicted class

label of weakly augmented example, the loss on unlabeled data can be formulated as Eq. 1:

$$\mathcal{L}_{unlab} = \sum_i \mathbb{1}(\max(c_i) \geq \tau)\mathcal{L}(\hat{c}_i, \widetilde{c}_i),\tag{1}$$

where $\mathcal{L}(\hat{c}_i, \widetilde{c}_i)$ is the loss between a class label and a confidence distribution.

Although FixMatch has become the foundation of many state-of-the-art SSL methods (Zhang et al., 2021; Zheng et al., 2022), we argue that it may fail to use unlabeled examples effectively and efficiently. (1) **Incorrect pseudo labels** caused by calibration error. A well-calibrated model is expected to be desirable if the predicted confidence score really reflects the probability of classifying the example correctly. However, according to Guo et al. (2017), most networks suffer from calibration error problems, such that the models become over-confident or under-confident. Hence, the confidence score cannot correctly indicate the chance that the example is correctly classified. The previous methods based on the confidence score can sometimes generate wrong pseudo labels, leading to the performance degeneration problem. Hence FixMatch-like methods appear to be unreliable. (2) **Huge computation cost** in training. The SSL model is required to forward propagate over the whole dataset to compute confidence scores for pseudo-label selection. Due to the great amount of unlabeled data, this step would be extremely time-consuming. However, not all unlabeled data is helpful to the model's decision boundary. For example, some data points can be too easy to provide meaningful gradients, while some can be too difficult for the model to select and learn at this stage. We argue that the unlabeled training set should be dynamically pruned, so as to reduce computation cost and speed up convergence.

To address the first issue, we propose **V**ariational **C**onfidence **C**alibration (**VCC**), a variational method to obtain the calibrated confidence scores for pseudo-label selection. The well-calibrated confidence score is expected to be closer to the ground-truth probability that an example is correctly predicted, providing a better reference in selecting pseudo-labeled examples. Although confidence calibration is a well-studied problem in fully-supervised setting, we argue it would be more challenging in SSL due to the absence of ground-truth labels. To bypass this difficulty, we employ three consistency scores to measure the stability of prediction. By simultaneously considering the stability and confidence of the prediction, we can approximate the calibrated confidence scores. Furthermore, the variational autoencoder is used to provide more stable results by reconstructing the calibrated confidences.

To address the second issue, we propose the **IN**fluence **F**unction-based **U**nlabeled **S**ample **E**limination (**INFUSE**) method. INFUSE uses the influence function Koh & Liang (2017) to compute the importance of each unlabeled example. By dynamically preserving the data points with the highest importance, the unlabeled core set can be built for replacing the whole dataset. On this small-scale core set, the model is expected to converge faster so that the computation cost at the training stage can be reduced. By combining two together, the finalized **VCC-INFUSE** method achieves higher prediction accuracy with lower training costs.

In summary, this paper makes the following contributions:

1. We propose the VCC method. By generating a well-calibrated confidence score, VCC can bring more accurate pseudo labels and improve the model's accuracy. As a pluggable module, VCC can be combined with existing SSL methods flexibly.

2. We propose the INFUSE method. INFUSE can dynamically prune unimportant unlabeled examples, in order to speed up the convergence and reduce the computation costs in training.

3. The effectiveness of our methods is demonstrated on multiple datasets and in various settings.

## 2 RELATED WORK

**Semi-Supervised Learning.** FixMatch (Sohn et al., 2020) is one of the most popular SSL methods. In FixMatch, the weakly-augmented unlabeled example is first fed to the model to obtain the one-hot pseudo-label. Then the model is trained with the strongly-augmented example and required to produce predictions consistent with the pseudo-label. FlexMatch (Zhang et al., 2021) further proposes an adaptive threshold strategy corresponding to the different learning stages and categories. SimMatch (Zheng et al., 2022) simultaneously considers semantic similarity and instance similarity, and encourages the same class predictions and similar similarity relationships for the same instance.

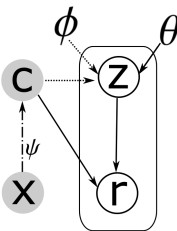

Figure 1: The graphical model of VCC. Here $x$ is the input, $c$ is the originally predicted confidence distribution, $z$ is the latent variable sampled from the encoder, and $r$ is the reconstructed confidence for pseudo label selection. Dash-dotted line denotes the original prediction function $p_\psi(c|x)$. Solid lines denote the generative model $p_\theta(r|c,z,x)$. Dashed lines denote the approximation $q_\phi(z|c,x)$ to the intractable posterior $p_\theta(z|c,r,x)$. The variational parameter $\phi$ is learned jointly with the generative model parameter $\theta$.

Apart from these methods, an explicit consistency regularization is also widely used (Laine & Aila, 2016; Berthelot et al., 2020; Miyato et al., 2019; Ganev & Aitchison, 2020; Chen et al., 2023; Li et al., 2021).

**Confidence Calibration.** Guo et al. (2017) is the first to point out the calibration problem in modern classifiers. They propose Temperature Scaling (TS) to rescale confidence distribution for preventing over-confident. Ensemble TS (Zhang et al., 2020) further extends the representation ability of TS by extending the parameter space. Besides, Kumar et al. (2018) propose the MMCE method, a trainable calibration regularization based on RKHS. However, these methods are restricted to the fully-supervised setting where the ground-truth label is available.

**Core Set Selection.** Most methods for selecting core set focus on the fully-supervised setting. Paul et al. (2021) propose the EL2N method, where the norm of the loss over an example is used to measure its importance. By keeping the most important examples, EL2N significantly reduces the training time at the cost of minor accuracy reduction. Killamsetty et al. (2021a) further propose GradMatch, which extended the core dataset to a weighted set by a submodular function. RETRIEVE (Killamsetty et al., 2021b) is most related to our work since it is designed for SSL. RETRIEVE formulates the core set selection as an optimizing problem. However, we argue that the optimizing function in RETRIEVE only considers the loss on the labeled training set, which may lead to a deviation from the desired results (i.e. minimizing the loss on the validation set).

## 3 CONFIDENCE CALIBRATION WITH VCC

Most existing calibration methods are not suitable for SSL due to the absence of ground-truth labels for unlabeled examples. Taking the original confidence score for pseudo-label selection will cause unstable results. Hence, we employ three different consistency scores ($s^{ens}$, $s^{tem}$ and $s^{view}$) to simultaneously measure the stability of prediction. By combing the three scores, we can obtain the approximated calibrated confidence $\tilde{r}$, which is closer to the probability of an example being correctly classified. However, $\tilde{r}$ is not directly used for pseudo-label selection since the process of estimating $\tilde{r}$ from three consistency scores is still unstable on some examples. Hence, we introduce VAE to reconstruct $\tilde{r}$ for selecting the pseudo-label. The graphical model and framework illustration of VCC is given by Fig. 1 and 2, respectively. The VAE is learned jointly with the original classifier in training, where $\tilde{r}$ is supposed to be the "ground-truth" to calculate the reconstruction loss. For selecting pseudo-label, we employ the output of VAE as the calibrated confidence.

### 3.1 ENSEMBLE CONSISTENCY

From the perspective of Bayesian, the parameters $\theta$ of a model are sampled from a probability distribution over the training set $D$. The model's prediction for a sample $x$ can be formulated as:

$$p(y|x, D) = \int p(y|x,\theta)p(\theta|D)\mathrm{d}\theta, \tag{2}$$

where $p(y|x,\theta)$ represents the probability distribution of the label $y$ of $x$ given the parameters $\theta$, and $p(\theta|D)$ represents the probability distribution of the model parameters $\theta$ trained on the dataset $D$. A single model may provide incorrect predictions for example $x$ due to randomness and noise, even if the confidence is high. Considering the entire parameter space, if all model parameters yield consistent predictions for $x$, the result is more convincing. In this case, the prediction can be viewed as an ensemble of predictions from multiple models.

However, due to the large parameter space of $\theta$, direct computation of Eq. 2 is intractable. In this study, we apply Monte-Carlo Dropout (Gal & Ghahramani, 2016) on the linear layer to approximate

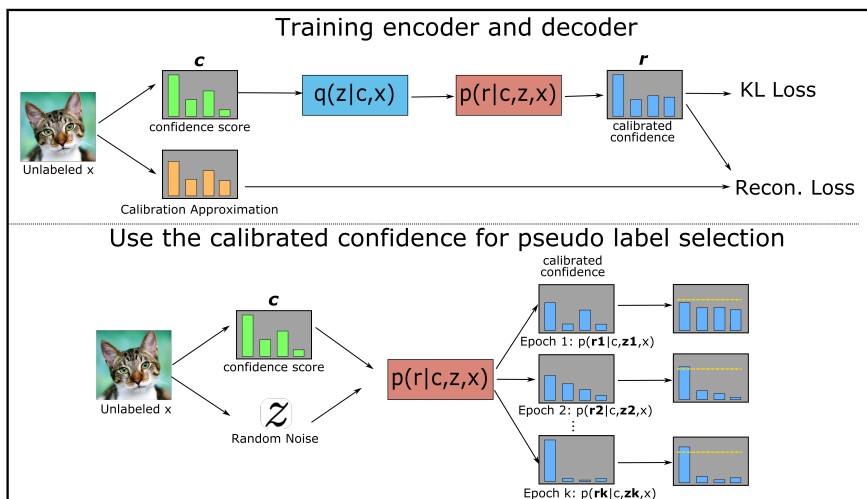

Figure 2: The illustration of VCC about training VAE and using the reconstructed confidence for pseudo-label selection.

the computation of Eq. 2. The feature map is cloned by $K$ copies, followed by a Dropout layer to randomly eliminate neural connections in the classification head to obtain predictions. By doing so, the model will generate $K$ estimated confidence distributions of example $i$, the expectation can be treated as the ensemble of $K$ different models:

$$\hat{\mathbf{y}}_i = p(y|x, \text{Dropout}(\theta)), \quad \tilde{\mathbf{y}} = \frac{1}{K}\sum_{i=1}^{K}\hat{\mathbf{y}}_i. \tag{3}$$

Then, entropy is employed as the ensemble-consistency score to measure the different models' consistency of example: $s^{ens} = -\sum_{c=1}^{M}\tilde{\mathbf{y}}_c \log \tilde{\mathbf{y}}_c$, where $M$ is the number of classes and $c$ is the index of the category.

## 3.2 TEMPORAL CONSISTENCY

In SSL, the model parameters are updated during training, causing the decision boundary to change frequently. Some examples may shift from one side of the decision boundary to the other after parameter updates, resulting in a change in classification results. In this case, even for examples with high confidence at the current step, their prediction results may be unstable. If these examples are used in training, it may result in incorrect pseudo labels and hinder the model's performance.

To measure the stability of prediction results between different stages, we propose the temporal consistency score, which considers the changes in confidence distribution of an example between different epochs. Specifically, let $y^t$ represent the confidence distribution of an example at epoch $t$. The temporal consistency score can be calculated as:

$$s^{tem} = D_{KL}\left(y^t \middle\| \frac{1}{K}\sum_{k=1}^{K}y^{t-k}\right) = \sum_{c=1}^{M}y_c^t \log\left(\frac{y_c^t}{\frac{1}{K}\sum_{k=1}^{K}y_c^{t-k}}\right), \tag{4}$$

where $D_{KL}$ represents the Kullback-Leibler Divergence, $M$ is the number of classes, and $K$ is a hyperparameter representing the window size. In experiments, we empirically set $K = 1$ to preserve the sensitivity of abnormal confidences. Although both consider the problem from the perspective of time, our temporal-consistency method is very dissimilar from the time-consistency method proposed by Zhou et al. (2020).

## 3.3 VIEW CONSISTENCY

Multi-view learning (Xu et al., 2015) aims to use multiple perspectives to predict data, so that different predictors can correct the prediction together. In SSL, to obtain models with different views, one

approach is to divide the whole dataset into multiple subsets for training multiple models. However, it incurs high model training costs. In the meanwhile, the volume of labeled data in each subset would be too small to train a decent model. To address this issue, we use Exponential Moving Average (EMA) to construct models with different views. The original model parameter $\theta$ is updated using gradient descent, while $\theta_{ema}$ is updated using the EMA scheme:

$$\theta_{ema}^t = \theta^t \cdot \beta + \theta_{ema}^{t-1} \cdot (1 - \beta), \tag{5}$$

where $\beta$ is a decay hyperparameter. Therefore, they can be treated as two different views from the same network structure. Typically, a classification model is composed of a feature extraction network (backbone) and a classification head (linear layer). To further increase the difference between the two views, we adopt a cross-feature trick. The backbone of each view first extracts features from the input, then fed into the classification head of the other view. It can be formulated as:

$$y = p(y|x, \theta^{backbone}, \theta_{ema}^{head}), \quad y_{ema} = p(y|x, \theta_{ema}^{backbone}, \theta^{head}). \tag{6}$$

After obtaining the outputs, the KL divergence is used to measure the consistency between them:

$$s^{view} = D_{KL}\left(y || y_{ema}\right). \tag{7}$$

It may seem like that temporal consistency and view consistency appear to overlap to some extent, as the predictions of the EMA model used in view consistency can also be considered an ensemble of predictions from past epochs. The difference is that the cross-feature trick is used at the computing of view consistency, which enforces this metric to focus more on consistency over multiple views rather than multiple time steps.

### 3.4    APPROXIMATION OF CALIBRATED CONFIDENCE

We have introduced three scores to evaluate the stability of prediction. However, $s^{ens}$, $s^{tem}$ and $s^{view}$ cannot be directly used for pseudo-label selection, which is based on confidence scores. To address this, we propose a simple method for approximating the calibrated confidence with the three consistency scores. The consistency scores are first normalized and summed up as the stability score. Then, we use interpolation to approximate the calibrated confidence $\tilde{r}_u$. Please refer to Appendix A for technical details.

### 3.5    RECONSTRUCT $\tilde{r}_u$ WITH VARIATIONAL AUTO ENCODER (VAE)

In Sec. 3.4, we combined three consistency scores to obtain $\tilde{r}_u$, which is the approximation of calibrated confidence scores. However, it may face instability due to the update of queue $q$ and abnormal interpolation endpoint (detail mentioned in Appendix A). To address this, we reconstruct the statistical-based $\tilde{r}_u$ in a learning-based way. Specifically, a VAE is employed to generate the calibrated confidence score $r_u$ for pseudo-label selection, and $\tilde{r}_u$ is used as input for training VAE.

We assume $r$ is generated by the following random process, which includes two steps: (1) a hidden variable $z$ which is sampled from a prior distribution $p_\theta(z)$; (2) a value $r$ is generated from the conditional distribution $p_\theta(r|c, z, x)$:

$$p_\theta(r|c, x) = \int_z p_\theta(z)p_\theta(r|z, c, x)\mathrm{d}z. \tag{8}$$

However, the marginal likelihood $p_\theta(r|c, x)$ is intractable generally. Hence another distribution $q_\phi(z|c, x)$ is introduced as the approximation of $p_\theta(z)$ (please refer to Appendix B for details):

$$\log p_\theta(r|c, x) = \int_z q_\phi(z|c, x) \log p_\theta(r|c, x)\mathrm{d}z$$
$$\geq \mathbb{E}_{q_\phi(z|c,x)} \log p_\theta(r|c, z, x) - D_{KL}(q_\phi(z|c, x)||p_\theta(z|c, x)). \tag{9}$$

The first term is the likelihood of calibration reconstruction (denoted as $\mathcal{L}_{VCC}^{recon}$), where $q_\phi(z|c, x)$ is the encoder to infer the hidden variable $z$, and $p_\theta(r|c, z, x)$ is the decoder to recover a calibrated confidence $r$. To compute the reconstruction loss, the approximated $\tilde{r}$ (Eq. 20 in Appendix A) is used as the ground truth. Besides, $z$ need to be sampled from $q_\phi(z|c, x)$. We use the reparameterization trick (Kingma & Welling, 2014) that uses the encoder to predict the mean and standard deviation of $z$.

By setting $\epsilon \sim \mathcal{N}(0,1)$, the reparameterization is formulated as $z = \mu(c,x) + \epsilon \cdot \sigma(c,x)$. As for the second term, under the Gaussian assumptions of the prior $p_\theta(z|c,x) \sim \mathcal{N}(0,1)$ and the approximator $q_\phi(z|c,x) \sim \mathcal{N}(\mu(c,x), \sigma^2(c,x))$, we have:

$$\mathcal{L}_{VCC}^{KL} \doteq D_{KL}(q_\phi(z|c,x) \| p_\theta(z|c,x)) = -\log\sigma + \frac{\mu^2 + \sigma^2}{2} - \frac{1}{2}. \tag{10}$$

The overall objective function for model training can be formulated as:

$$\mathcal{L} = \mathcal{L}_{lab} + \lambda_{unlab} \cdot \mathcal{L}_{unlab} + \lambda_{VCC} \cdot \left( \mathcal{L}_{VCC}^{recon} - \mathcal{L}_{VCC}^{KL} \right). \tag{11}$$

Although we generate more accurate confidence score by combining three consistencies, this confidence score is still not as optimal as the inaccessible ground-truth. This is because there are many other "nuisance" and untraceable factors that affect the approaching of pseudo label towards the ground-truth, such as the randomness of the neural networks. Under this circumstance, directly approaching the unreliable target may still degrade the performance. The original VAE is proposed to learn continuous distribution features from discontinuous distributions by sampling a hidden variable. This process is suitable for suboptimal pseudo label learning, because the approaching of the prediction to the generated pseudo label can be viewed as the process of the approaching of the prediction to the ground-truth. Since eliminating those nuisance factors cannot be tractable, we use VAE to simulate this process instead of the MLP.

## 4 CORE SET SELECTION WITH INFUSE

In the previous section, we introduce the VCC framework, which ensures well-calibrated confidence scores to improve the accuracy in pseudo-label selection. Nonetheless, as we previously discussed, training the SSL model still encounters substantial computational expenses. Furthermore, the incorporation of the additional encoder and decoder of the VCC introduces an extra computation overhead. To address these challenges, we present INFUSE—a core set selection methodology aimed at efficient example selection. Based on influence function Koh & Liang (2017), INFUSE allows for training the SSL model by using only a subset of the complete unlabeled dataset, so that the training time can be significantly reduced.

In SSL, the model should minimize the loss on the validation set to obtain the highest generalization accuracy:

$$\min \mathcal{L}(V, \theta^*), \quad \text{s.t.} \ \ \theta^* = \arg\min_\theta \ R(\theta), \tag{12}$$

$$R(\theta) \doteq \mathbb{E}_{(x,y) \in S} \left[ H(q_x, y) \right] + \lambda \cdot \mathbb{E}_{u \in U} \left[ \mathbb{1} \left( \max(q_u) \geq \tau \right) \cdot H\left( \hat{q}_u, p\left( y \mid u \right) \right) \right].$$

Here $H$ is the loss function, $\tau$ is the threshold for pseudo label selection, $q$ is the confidence distribution, $\hat{q}$ is the pseudo label, and $R(\theta)$ is the total loss on labeled dataset $S$ and unlabeled dataset $U$. Now assume the weight of an unlabeled example $u'$ is increased by $\epsilon$. Denote $\mathcal{L}_U(u', \theta) = \lambda \cdot \mathbb{1} \left( \max(q_{u'}) \geq \tau \right) \cdot H\left( \hat{q}_{u'}, p\left( y \mid u' \right) \right)$, the optimal model parameters corresponding to the new training set become:

$$\hat{\theta} = \arg\min_\theta \ R(\theta) + \epsilon \cdot \mathcal{L}_U(u', \theta). \tag{13}$$

In Eq. 13, $\hat{\theta}$ minimizes the loss function on the training set, which means the gradient w.r.t $\hat{\theta}$ is 0:

$$\nabla_\theta R(\hat{\theta}) + \epsilon \nabla_\theta \mathcal{L}_U(u', \hat{\theta}) = 0. \tag{14}$$

Using a Taylor-series approximation at $\theta^*$, Eq. 14 can be rewritten as:

$$\nabla_\theta R\left(\theta^*\right) + \epsilon \cdot \nabla_\theta \mathcal{L}_U\left(u', \theta^*\right) + \left( \nabla_\theta^2 R\left(\theta^*\right) + \epsilon \cdot \nabla_\theta^2 \mathcal{L}_U\left(u', \theta^*\right) \right) \cdot \left( \hat{\theta} - \theta^* \right) = 0, \tag{15}$$

which gives (please refer to Appendix C for details):

$$\hat{\theta} - \theta^* \approx - \left( \nabla_\theta^2 R\left(\theta^*\right) \right)^{-1} \cdot \epsilon \nabla_\theta \mathcal{L}_U\left(u', \theta^*\right) \doteq -\epsilon \cdot H_\theta^{-1} \nabla_\theta \mathcal{L}_U\left(u, \theta\right). \tag{16}$$

With the help of chain rule $\frac{d\mathcal{L}}{d\epsilon} = \frac{d\mathcal{L}}{d\theta} \cdot \frac{d\theta}{d\epsilon}$, the importance of an unlabeled example can be estimated:

$$\text{score}_\theta(u) = \frac{d\mathcal{L}(V, \theta)}{d\epsilon} = \nabla_\theta \mathcal{L}(V, \theta)^\top \frac{d\theta}{d\epsilon} = -\nabla_\theta \mathcal{L}(V, \theta)^\top H_\theta^{-1} \nabla_\theta \mathcal{L}_U\left(u, \theta\right). \tag{17}$$

| Method | CIFAR-10 | | | CIFAR-100 | | | SVHN | | |
|---|---|---|---|---|---|---|---|---|---|
| | 40 | 250 | 2500 | 400 | 2500 | 10000 | 40 | 250 | 1000 |
| PL | $76.29_{\pm1.08}$ | $48.28_{\pm2.01}$ | $14.90_{\pm0.20}$ | $87.15_{\pm0.47}$ | $59.09_{\pm0.61}$ | $38.86_{\pm0.09}$ | $75.95_{\pm3.39}$ | $16.60_{\pm1.13}$ | $9.33_{\pm0.58}$ |
| UDA | $8.01_{\pm1.34}$ | $5.12_{\pm0.15}$ | $4.32_{\pm0.07}$ | $53.44_{\pm2.06}$ | $34.37_{\pm0.28}$ | $27.52_{\pm0.10}$ | $2.03_{\pm0.02}$ | $2.03_{\pm0.03}$ | $1.96_{\pm0.01}$ |
| VAT | $76.42_{\pm2.57}$ | $42.58_{\pm6.67}$ | $10.97_{\pm0.19}$ | $83.11_{\pm0.27}$ | $53.17_{\pm0.57}$ | $36.58_{\pm0.21}$ | $77.00_{\pm6.59}$ | $4.59_{\pm0.13}$ | $4.09_{\pm0.21}$ |
| MeanTeacher | $76.93_{\pm2.29}$ | $56.06_{\pm2.03}$ | $15.47_{\pm0.43}$ | $90.34_{\pm0.65}$ | $61.13_{\pm0.57}$ | $39.05_{\pm0.12}$ | $81.94_{\pm1.33}$ | $25.10_{\pm3.17}$ | $12.29_{\pm0.45}$ |
| MixMatch | $70.67_{\pm1.25}$ | $37.28_{\pm0.61}$ | $7.38_{\pm0.06}$ | $79.95_{\pm0.29}$ | $49.58_{\pm0.62}$ | $32.10_{\pm0.13}$ | $79.63_{\pm5.78}$ | $3.71_{\pm0.20}$ | $3.12_{\pm0.09}$ |
| ReMixMatch | $14.50_{\pm2.58}$ | $9.21_{\pm0.55}$ | $4.89_{\pm0.05}$ | $57.10_{\pm0.01}$ | $34.77_{\pm0.32}$ | $26.18_{\pm0.23}$ | $31.27_{\pm18.79}$ | $6.38_{\pm1.09}$ | $5.34_{\pm0.45}$ |
| Dash(RandAug) | $15.01_{\pm3.70}$ | $5.13_{\pm0.26}$ | $4.35_{\pm0.09}$ | $53.98_{\pm2.31}$ | $34.47_{\pm0.12}$ | $27.72_{\pm0.03}$ | $2.08_{\pm0.09}$ | $1.97_{\pm0.01}$ | $2.03_{\pm0.03}$ |
| SoftMatch | $5.06_{\pm0.02}$ | $4.84_{\pm0.10}$ | $4.27_{\pm0.12}$ | $49.64_{\pm1.46}$ | $33.05_{\pm0.05}$ | $27.26_{\pm0.03}$ | $2.31_{\pm0.01}$ | $2.15_{\pm0.05}$ | $2.08_{\pm0.04}$ |
| CoMatch | $5.44_{\pm0.05}$ | $5.33_{\pm0.12}$ | $4.29_{\pm0.04}$ | $60.98_{\pm0.77}$ | $37.24_{\pm0.24}$ | $28.15_{\pm0.16}$ | $9.51_{\pm5.59}$ | $2.21_{\pm0.20}$ | $1.96_{\pm0.07}$ |
| FixMatch | $7.52_{\pm0.42}$ | $4.90_{\pm0.03}$ | $4.28_{\pm0.10}$ | $46.47_{\pm0.05}$ | $28.09_{\pm0.06}$ | $22.21_{\pm0.02}$ | $\mathbf{2.96}_{\pm1.23}$ | $1.99_{\pm0.05}$ | $1.96_{\pm0.06}$ |
| VCC-FixMatch | $\mathbf{6.84}_{\pm0.52}$ | $\mathbf{4.68}_{\pm0.04}$ | $\mathbf{4.27}_{\pm0.21}$ | $\mathbf{43.31}_{\pm0.02}$ | $\mathbf{27.76}_{\pm0.06}$ | $\mathbf{22.05}_{\pm0.03}$ | $3.12_{\pm0.61}$ | $\mathbf{1.97}_{\pm0.02}$ | $\mathbf{1.95}_{\pm0.08}$ |
| FlexMatch | $4.98_{\pm0.01}$ | $5.00_{\pm0.05}$ | $4.24_{\pm0.07}$ | $40.43_{\pm0.63}$ | $26.38_{\pm0.17}$ | $21.83_{\pm0.08}$ | $3.36_{\pm0.37}$ | $5.02_{\pm1.20}$ | $5.43_{\pm0.46}$ |
| VCC-FlexMatch | $\mathbf{4.90}_{\pm0.10}$ | $\mathbf{4.65}_{\pm0.07}$ | $\mathbf{4.14}_{\pm0.15}$ | $\mathbf{37.98}_{\pm0.65}$ | $\mathbf{25.75}_{\pm0.11}$ | $\mathbf{21.48}_{\pm0.07}$ | $\mathbf{2.62}_{\pm0.08}$ | $\mathbf{4.97}_{\pm0.08}$ | $\mathbf{3.71}_{\pm1.13}$ |
| SimMatch | $5.60_{\pm1.37}$ | $4.84_{\pm0.39}$ | $3.96_{\pm0.01}$ | $37.81_{\pm2.21}$ | $25.07_{\pm0.32}$ | $\mathbf{20.58}_{\pm0.11}$ | $3.70_{\pm0.72}$ | $2.27_{\pm0.12}$ | $\mathbf{2.07}_{\pm0.08}$ |
| VCC-SimMatch | $\mathbf{5.27}_{\pm0.34}$ | $\mathbf{4.76}_{\pm0.14}$ | $\mathbf{3.87}_{\pm0.24}$ | $\mathbf{37.22}_{\pm0.04}$ | $\mathbf{24.98}_{\pm0.13}$ | $20.61_{\pm0.01}$ | $\mathbf{3.04}_{\pm0.02}$ | $\mathbf{2.20}_{\pm0.01}$ | $4.39_{\pm0.02}$ |
| Fully-Supervised | | $4.58_{\pm0.05}$ | | | $19.63_{\pm0.08}$ | | | $2.07_{\pm0.02}$ | |

Table 1: Comparison of error rate (%) for different methods under various settings.

| Labels | MeanTeacher | MixMatch | ReMixMatch | FixMatch | w/ VCC | FlexMatch | w/ VCC | SimMatch | w/ VCC |
|---|---|---|---|---|---|---|---|---|---|
| 40 | 71.72 | 54.93 | 32.12 | 35.97 | $\mathbf{30.63}(\downarrow 5.34)$ | 29.15 | $\mathbf{28.14}(\downarrow 1.01)$ | 27.84 | $\mathbf{26.97}(\downarrow 0.87)$ |
| 1000 | 33.90 | 21.70 | 6.74 | 6.25 | $\mathbf{5.31}(\downarrow 0.94)$ | 5.77 | $\mathbf{5.52}(\downarrow 0.25)$ | 5.91 | $\mathbf{5.51}(\downarrow 0.40)$ |

Table 2: The error rate results (%) of different methods on STL-10 dataset.

| Method | 400 label | | | | 2500 label | | | | 10000 label | | | |
|---|---|---|---|---|---|---|---|---|---|---|---|---|
| | ER(%) | ECE | MCE | ACE | ER(%) | ECE | MCE | ACE | ER(%) | ECE | MCE | ACE |
| FixMatch | 46.42 | 0.382 | 0.573 | 0.376 | 28.03 | 0.208 | 0.530 | 0.199 | 22.20 | 0.127 | 0.322 | 0.128 |
| VCC-FixMatch | **43.29** | **0.359** | **0.560** | **0.345** | **27.81** | **0.195** | **0.418** | **0.182** | **22.01** | **0.125** | **0.317** | **0.127** |
| FlexMatch | 39.94 | 0.291 | 0.512 | 0.286 | 26.49 | 0.169 | 0.369 | 0.173 | 21.90 | 0.120 | 0.311 | 0.126 |
| VCC-FlexMatch | **37.52** | **0.257** | **0.446** | **0.258** | **25.26** | **0.147** | **0.324** | **0.163** | **21.55** | **0.104** | **0.269** | **0.125** |
| SimMatch | 37.81 | 0.325 | **0.510** | 0.328 | 25.07 | 0.157 | 0.358 | 0.179 | **20.58** | 0.113 | 0.295 | **0.116** |
| VCC-SimMatch | **37.20** | **0.317** | 0.514 | **0.314** | **25.01** | **0.155** | **0.347** | **0.173** | 20.61 | 0.115 | **0.291** | 0.121 |

Table 3: The error rate, ECE (Guo et al., 2017), MCE (Guo et al., 2017) and ACE (Nixon et al., 2019) results of different methods on CIFAR-100 dataset with 400/2500/10000 labeled examples.

Eq. 17 is used to compute $\text{score}_\theta(u)$ for each unlabeled example. The unlabeled examples with the highest score are preserved to build the core set and others will be simply dropped. In our implementation, the INFUSE score is calculated batch-wise to reduce the computation overhead. Besides, we use the identity matrix to approximate the inverse Hessian $H_\theta^{-1}$ (Luketina et al., 2016) for efficiency. The last problem is how to compute $\nabla_\theta \mathcal{L}(V, \theta)$ when the ground-truth label of examples in $V$ is unavailable in training. To address this, we propose a feature-level mixup to build a support set $\overline{S}$. Then, the gradient on the validation set is approximated by $\mathcal{L}(\overline{S}, \theta)$. Please refer to Appendix D for details.

## 5 EXPERIMENTS

The effectiveness of our method is evaluated on standard SSL datasets: CIFAR10/100 (Krizhevsky et al., 2009), SVHN (Netzer et al., 2011), STL-10 (Coates et al., 2011). We follow the most commonly used SSL setting (Sohn et al., 2020) to train the model (please refer to Appendix E for more details). Specifically, the keep ratio $k$ controls the size of core set. Taking $k = 10\%$ for example, the amount of examples in core set is $10\% \times |U|$, and the total steps also become $10\%$ of the original iterations.

### 5.1 MAIN RESULTS

In this section, we demonstrate the effectiveness of VCC and INFUSE separately, then combine them to achieve more efficient and accurate pseudo-label selection in SSL.

As mentioned before, VCC is a general confidence calibration plugin, which makes it possible to combine VCC with existing SSL methods flexibly. In experiments, we choose the popular FixMatch (Sohn et al., 2020), FlexMatch (Zhang et al., 2021), and SimMatch (Zheng et al., 2022) as the

| Method | CIFAR-10 | | | | | CIFAR-100 | | | | | STL-10 | | SVHN | |
|---|---|---|---|---|---|---|---|---|---|---|---|---|---|---|
| | 250 label | | | 4000 label | | 2500 label | | | 10000 label | | 250 label | | 250 label | |
| | 10% | 20% | 40% | 40% | 60% | 10% | 20% | 40% | 40% | 60% | 10% | 20% | 10% | 20% |
| Random | 9.12 | 6.87 | 6.51 | 5.26 | 5.01 | 31.55 | 31.11 | 28.86 | 23.19 | 22.51 | 16.62 | 14.37 | 3.85 | 4.65 |
| Earlystop | 7.47 | 6.03 | 6.85 | 4.86 | 4.52 | 29.21 | 28.85 | 27.30 | 23.03 | 22.61 | 16.31 | 13.20 | 2.93 | 3.08 |
| EL2N | 8.55 | 7.47 | 6.70 | 4.94 | 4.54 | 31.55 | 31.27 | 28.42 | 23.12 | 22.21 | 16.27 | 12.92 | 3.66 | 3.61 |
| GradMatch | 6.71 | 5.87 | 5.60 | 4.72 | 4.45 | 28.95 | 28.48 | 26.71 | 22.72 | 22.21 | 16.05 | 12.90 | 2.90 | 2.63 |
| RETRIEVE | 6.60 | 6.02 | 5.48 | 4.68 | 4.41 | **28.75** | 28.34 | 26.68 | 22.56 | 22.18 | 16.05 | 12.90 | 2.90 | 2.63 |
| INFUSE (Ours) | **6.29** | **5.69** | **5.33** | **4.51** | **4.34** | 28.83 | **28.05** | **26.47** | **22.28** | **21.97** | **15.84** | **12.71** | **2.61** | **2.46** |
| Full Unlabeled Data | 4.98 | | | 4.19 | | 26.49 | | | 21.90 | | 8.23 | | 3.80 | |

Table 4: Comparison of error rate (%) for core set selection methods on different datasets with varying example keep raito (from 10% to 60%).

| Method | Error Rate (%) | Training time (GPU Hours) |
|---|---|---|
| Dash(RandAug) | 27.15 | - |
| MPL | 27.71 | - |
| FixMatch | 28.03 | 221.91 |
| FlexMatch | 26.49 | 223.96 |
| VCC-FlexMatch (Ours) | **25.26** | 253.53 |
| VCC-INFUSE-FlexMatch (Ours, keep raito=40%) | 25.41 | **115.47** |

Table 5: The error rate and training time of different methods on CIFAR-100 dataset with 2500 labeled data. The GPU Hours metric is calculated based on the A100 GPU.

basic module to build VCC-FixMatch, VCC-FlexMatch, and VCC-SimMatch. We report the mean value and the standard deviation of three random independent trials of each setting, with the results shown in Table 1. All three baseline methods (FixMatch, FlexMatch, SimMatch) achieve accuracy improvements when combined with VCC for confidence calibration. Specifically, the improvements of VCC is more significant in the case that the amount of labeled examples is small. Taking the results on CIFAR-100 as an example, when only 400 labeled examples are available, VCC-FlexMatch reduces the error rate of FlexMatch from 46.47% to 43.31% (-3.16%). Similar boost is also produced when running on STL-10 dataset as shown in Table 2, where VCC reduces the error rate of FixMatch by 5.34% (from 35.97% to 30.63%) with only 40 labels.

To further verify the source of the accuracy improvement of VCC, we calculate the calibration error of different methods. As shown in Table 3, both VCC-FixMatch and VCC-FlexMatch achieve lower calibration errors compared to the baseline methods under various settings. VCC-SimMatch also achieves lower ECE and ACE metrics when only 400 labeled examples are available. However, the MCE metric is deteriorated, which is attributed to the fact that MCE considers the worst-calibrated bucket and introduces some fluctuations. Under the setting of using 10,000 labeled examples, the results of VCC-SimMatch and SimMatch are very close. This is partly because a larger number of labeled examples can naturally improve the model's performance and reduces the calibration error. Besides, SimMatch has employed instance similarity for rescaling the confidence score, which may reduce the benefits brought by VCC.

The results of INFUSE and other core set selection methods (e.g. RETRIEVE (Killamsetty et al., 2021b)) are shown in Table 4. On the CIFAR-10 dataset, INFUSE achieves a relatively low error rate (6.29%) using only 10% of the examples, indicating the original unlabeled data is redundant and proving the significance of core set selection in SSL. With the increase of the keep ratio, the gap between INFUSE and the non-pruned setting becomes smaller. For example, on the CIFAR-100 dataset when the amount of labeled data is 2500 and the keep ratio is 40%, INFUSE achieves an error rate of 26.47% while the baseline is 26.49%. When compared with other core set selection methods, INFUSE also achieves lower error rates in most settings.

The results above show the effectiveness of VCC and INFUSE respectively. By combining two together, we propose the VCC-INFUSE method. The results are shown in Table 5. VCC-INFUSE achieves a better trade-off between model performance and computation costs. Compared to Flex-Match, VCC-INFUSE-FlexMatch can not only reduce the error rate from 26.49% to 25.41% (-1.08%), but also reduce the training time from 223.96 GPU Hours to 115.47 GPU Hours (-48.44%).

| Method | Error Rate (%) |
|---|---|
| FixMatch (Sohn et al., 2020) | 25.07 |
| FlexMatch (Zhang et al., 2021) | 25.87 |
| SimMatch (Zheng et al., 2022) | 61.54 |
| FixMatch+DASO (Oh et al., 2022) | 24.63 |
| FixMatch+DebiasPL (Wang et al., 2022) | 24.42 |
| FixMatch+DARP (Kim et al., 2020) | 22.93 |
| FixMatch+Adsh (Guo & Li, 2022) | 21.88 |
| FixMatch+VCC (Ours) | **21.16** |

Table 6: The error rate (%) of different methods on CIFAR-10-LT under the class imbalance setting.

| Reconstruct $\tilde{r}_u$ by VAE | ER(%) | ECE | MCE | ACE |
|---|---|---|---|---|
| ✗ | 25.76 | 0.160 | 0.411 | 0.168 |
| ✓ | **25.26** | **0.147** | **0.324** | **0.163** |

Table 7: The error rate of VCC with or without reconstructing calibrated confidence on CIFAR-100 dataset with 2500 labeled examples.

## 5.2 SUPPLEMENTARY RESULTS

**Class Imbalance SSL.** We also design the experiments for more realistic settings such as class imbalance. The experiment results are shown in Table 6 (please refer to Appendix G for experiments settings). While FixMatch surpasses FlexMatch by $0.8\%$ with an error rate of $25.07\%$, SimMatch only achieves $61.54\%$, which shows a total failure on this task. DASO and DebiasPL slightly reduce the error rate to $24.63\%$ and $24.42\%$, respectively. DARP achieves better performance with an error rate of $22.93\%$. However, the proposed VCC, which is not designed for imbalance-SSL specifically, produces the lowest error rate of $21.16\%$, which is $0.72\%$ lower than the second-best method Adsh. The results further prove our method's ability to reduce bias and bring a more accurate pseudo-label.

**The Effectiveness of Reconstructing Calibrated Confidence by VAE.** In VCC, we first approximate the calibrated confidence to obtain $\tilde{r}_u$, then use VAE to reconstruct it to obtain $r_u$, which will be used in pseudo-label selection. The objective of reconstruction aims at alleviating the randomness of statistical approximation. To demonstrate the necessity, we conduct the ablation study. As shown in Table 7, VCC with reconstruction further reduces the error rate by 0.50%.

**VCC v.s. other calibration methods.** Although most calibration methods for the fully-supervised setting are unsuitable in SSL, the pseudo-label can be used to approximate ground truth. We choose Ensemble-TS (Zhang et al., 2020) and MMCE (Kumar et al., 2018) as the baseline to compare with VCC. As shown in Table 8 (Appendix F), the error rate of MMCE is the highest (28.44%). The reason is that MMCE directly uses the pseudo-label to calculate the calibration regularization, while the incorrect pseudo-label may bring the noise. As for Ensemble-TS, it uses pseudo-label to search the optimal parameter scaling, which can alleviate the problem of incorrect pseudo-label to some extent (ER=26.36%). As a comparison, VCC achieves the lowest error rate (25.26%) and the best calibration performance.

**The Ablation Study of Three Consistency Scores.** We use view consistency, temporal consistency, and ensemble consistency for estimating $\tilde{r}$. The three consistency scores are designed to reflect the stability of prediction from different perspectives. To analyze their contribution, we conduct the ablation study (Table 9 in Appendix F). As we can see, each consistency score contributes to the estimation of a more accurate $\tilde{r}$ so that a lower error rate can be achieved.

## 6 CONCLUSION

In this paper, we addressed the challenges of leveraging large-scale unlabeled data in SSL and proposed two novel methods, VCC and INFUSE, to improve the effectiveness and efficiency of data selection. As a general plugin, VCC significantly improves the accuracy of FixMatch, FlexMatch and SimMatch on multiple datasets. Simultaneously, INFUSE achieves competitive or even lower error rates with partial unlabeled data. By combining two together, VCC-INFUSE achieves a lower error rate with less computation overhead. The future work is to extend VCC-INFUSE to more SSL tasks (e.g. object detection, segmentation) to verify its generalization.

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

# Appendices

## A   APPROXIMATION OF CALIBRATED CONFIDENCE

First, a fixed-length queue $q$ is maintained to record the historical predictions of the unlabeled samples in mini-batches. Since $s^{ens}$, $s^{tem}$, and $s^{view}$ have different distributions, directly summing them is unfair. Therefore, they are normalized using max-min normalization. Let $u$ be the unlabeled example, the normalization is done as follows:

$$\tilde{s}_u^t = \frac{s_u^t - \min_{u' \in q}\left(s_{u'}^t\right)}{\max_{u' \in q}\left(s_{u'}^t\right) - \min_{u' \in q}\left(s_{u'}^t\right)}, \tag{18}$$

where $t = \{ens, tem, view\}$. After normalization, $\tilde{s}_u^{ens}$, $\tilde{s}_u^{tem}$, and $\tilde{s}_u^{view}$ are all real numbers ranging from 0 to 1. These consistency scores evaluate the stability of the examples. However, some hard-to-learn examples may also have stable predictions but low confidence scores. Hence, the three consistency scores are not enough to describe the reliability of the prediction. To address this, the original confidence score of the sample $\tilde{s}_u^{conf} = \max(y_u)$ is also used. Thus, an unlabeled sample $u$ can be represented by a quadruple $(\tilde{s}_u^{ens}, \tilde{s}_u^{tem}, \tilde{s}_u^{view}, \tilde{s}_u^{conf})$.

The next problem is how to combine these four scores together for estimation. To avoid complex parameter tuning, VCC adopts a simple yet effective approach: taking the sum of their squares:

$$s_u = \left(\tilde{s}_u^{ens}\right)^2 + \left(\tilde{s}_u^{tem}\right)^2 + \left(\tilde{s}_u^{view}\right)^2 + \left(\tilde{s}_u^{conf}\right)^2. \tag{19}$$

According to the results in Guo et al. (2017), calibration errors mainly occur in the middle range of confidences, while samples with extremely low or high confidences tend to have smaller calibration errors. Therefore, we approximately treat the lowest/highest confidence score in $q$ as well-calibrated and employs interpolation to calculate the calibrated confidence scores for other examples. To further eliminate the unfairness between different categories, the interpolation operation only considers examples with the same pseudo labels as the current example $u$.

$$q' = \{e \mid e \in q, \arg\max \tilde{s}_e^{conf} = \arg\max \tilde{s}_u^{conf}\},$$
$$max\_score = \max_{u' \in q'}(s_{u'}), \qquad min\_score = \min_{u' \in q'}(s_{u'}),$$
$$max\_conf = \max_{u' \in q'}\left(\tilde{s}_{u'}^{conf}\right), \quad max\_conf = \max_{u' \in q'}\left(\tilde{s}_{u'}^{conf}\right), \tag{20}$$
$$\tilde{r}_u = \frac{max\_score - s_u}{max\_score - min\_score} \cdot (max\_conf - min\_conf) + min\_conf.$$

## B   OPTIMIZING VAE IN VCC

In Eq. 9, we use another distribution $q_\phi(z|c, x)$ as the approximation of $p_\theta(z)$:

$$\log p_\theta(r|c, x)$$
$$= \int_z q_\phi(z|c, x) \log p_\theta(r|c, x) \mathrm{d}z$$
$$= \int_z q_\phi(z|c, x) \log \frac{p_\theta(r|c, z, x) p_\theta(z|c, x)}{p_\theta(z|r, c, x)} \mathrm{d}z$$
$$= \int_z q_\phi(z|c, x) \log \left(\frac{p_\theta(r|c, z, x) p_\theta(z|c, x)}{p_\theta(z|r, c, x)} \frac{q_\phi(z|c, x)}{q_\phi(z|c, x)}\right) \mathrm{d}z$$
$$= \int_z q_\phi(z|c, x) \left(\log \frac{p_\theta(r|c, z, x) p_\theta(z|c, x)}{q_\phi(z|c, x)} + \log \frac{q_\phi(z|c, x)}{p_\theta(z|r, c, x)}\right) \mathrm{d}z$$
$$= \int_z q_\phi(z|c, x) \log \frac{p_\theta(r|c, z, x) p_\theta(z|c, x)}{q_\phi(z|c, x)} \mathrm{d}z + D_{KL}(q_\phi(z|c, x) \| p_\theta(z|c, r, x))$$
$$\geq \int_z q_\phi(z|c, x) \log \frac{p_\theta(r|c, z, x) p_\theta(z|c, x)}{q_\phi(z|c, x)} \mathrm{d}z, \tag{21}$$

where the second equation employs the Bayes' theorem: $p(r) = p(r, z)/p(z|r) = p(r|z)p(z)/p(z|r)$. In Inequality 9, the non-negative Kullback-Leibler divergence $D_{KL}(q\|p)$ cannot be directly computed and the remaining part is called the Evidence Lower Bound (ELBO) of variational. To find the optimal $q_\phi(z|c, x)$ to approximate $p_\theta(z|c, r, x)$, the ELBO requires to be maximized. The Inequality 9 can be further rewritten as:

$$\log p_\theta(r|c, x) \geq \int_z q_\phi(z|c, x) \log \frac{p_\theta(r|c, z, x)p_\theta(z|c, x)}{q_\phi(z|c, x)} dz$$

$$= \mathbb{E}_{q_\phi(z|c,x)} \log p_\theta(r|c, z, x) - D_{KL}(q_\phi(z|c, x)\|p_\theta(z|c, x)). \tag{22}$$

The first term is the likelihood of calibration reconstruction, where $q_\phi(z|c, x)$ is the encoder to infer the hidden variable $z$, and $p_\theta(r|c, z, x)$ is the decoder to recover a calibrated confidence $r$. Under the Gaussian assumptions of the prior $p_\theta(z|c, x) \sim \mathcal{N}(0, 1)$ and the approximator $q_\phi(z|c, x) \sim \mathcal{N}(\mu(c, x), \sigma^2(c, x))$, the second term is equal to:

$$D_{KL}(q_\phi(z|c, x)\|p_\theta(z|c, x))$$

$$= \int_z q_\phi(z|c, x)(\log q_\phi(z|c, x) - \log p_\theta(z|c, x)) dz$$

$$= \int_z q_\phi(z|c, x)(\log \frac{1}{\sqrt{2\pi\sigma^2}} e^{-\frac{(z-\mu)^2}{2\sigma^2}} - \log \frac{1}{\sqrt{2\pi}} e^{-\frac{z^2}{2}}) dz$$

$$= \int_z q_\phi(z|c, x)\left(-\frac{(z-\mu)^2}{2\sigma^2} + \log \frac{1}{\sqrt{2\pi\sigma^2}} + \frac{z^2}{2} - \log \frac{1}{\sqrt{2\pi}}\right) dz$$

$$= -\int_z q_\phi(z|c, x) \log \sigma dz + \int_z q_\phi(z|c, x)\frac{z^2}{2} dz - \int_z q_\phi(z|c, x)\frac{(z-\mu)^2}{2\sigma^2} dz \tag{23}$$

$$= -\log \sigma + \mathbb{E}_{z \sim q_\phi}\left[\frac{z^2}{2}\right] - \mathbb{E}_{z \sim q_\phi}\left[\frac{(z-\mu)^2}{2\sigma^2}\right]$$

$$= -\log \sigma + \frac{1}{2}\left((\mathbb{E}_{z \sim q_\phi}[z])^2 + Var(z)\right) - \frac{1}{2\sigma^2}Var(z)$$

$$= -\log \sigma + \frac{\mu^2 + \sigma^2}{2} - \frac{1}{2},$$

where the second last equation employs the variance lemma: $\mathbb{E}\left[z^2\right] = (\mathbb{E}\left[z\right])^2 + Var(z)$. To compute the reconstruction error, the $z$ need to be sampled from $q_\phi(z|c, x)$. We use the reparameterization Kingma & Welling (2014) trick to address this. By setting $\epsilon \sim \mathcal{N}(0, 1)$, the reparameterization is formulated as $z = \mu(c, x) + \epsilon \cdot \sigma(c, x)$.

## C    THE DETAILED DEDUCTION OF INFUSE

To deduce Eq. 16 from Eq. 15, we have:

$$\hat{\theta} - \theta^* = -\left[\nabla_\theta^2 R\left(\theta^*\right) + \epsilon \cdot \nabla_\theta^2 \mathcal{L}_U\left(u', \theta^*\right)\right]^{-1}\left[\nabla_\theta R\left(\theta^*\right) + \epsilon \cdot \nabla_\theta \mathcal{L}_U\left(u', \theta^*\right)\right]. \tag{24}$$

Here $\nabla_\theta R\left(\theta^*\right) = 0$, since $\theta^*$ is the optimal parameters that minimize $R(\theta)$:

$$\hat{\theta} - \theta^* = -\left[\nabla_\theta^2 R\left(\theta^*\right) + \epsilon \cdot \nabla_\theta^2 \mathcal{L}_U\left(u', \theta^*\right)\right]^{-1} \cdot \epsilon \nabla_\theta \mathcal{L}_U\left(u', \theta^*\right). \tag{25}$$

Note that $\epsilon$ is an extremely small number. For the entity of $\left[\nabla_\theta^2 R\left(\theta^*\right) + \epsilon \cdot \nabla_\theta^2 \mathcal{L}_U\left(u', \theta^*\right)\right]^{-1}$, the contribution of $\epsilon \cdot \nabla_\theta^2 \mathcal{L}_U\left(u', \theta^*\right)$ is so small that we can approximately omit it. Now we have:

$$\hat{\theta} - \theta^* = -\epsilon \left(\nabla_\theta^2 R\left(\theta^*\right)\right)^{-1} \nabla_\theta \mathcal{L}_U\left(u', \theta^*\right). \tag{26}$$

## D    APPROXIMATE $\nabla_\theta \mathcal{L}\left(V, \theta\right)$ WITH MIXUP

In Eq. 17, the first term $\nabla_\theta \mathcal{L}\left(V, \theta\right)^\top$ is the gradient on the validation set. However, it's infeasible to direct compute since the ground-truth label of examples in $V$ is unavailable in training. One

applicable approximation is $\nabla_\theta \mathcal{L}(V, \theta) \approx \nabla_\theta \mathcal{L}(S, \theta)$, i.e. use the gradient on labeled training set. However, the volume of the labeled dataset is small in SSL and the model tends to overfit quickly on that. Hence, $\nabla_\theta \mathcal{L}(S, \theta)$ may bring huge error in practice. Another way is to approximate $\nabla_\theta \mathcal{L}(V, \theta)$ with the gradient of unlabeled dataset. However, the pseudo label can be noisy (especially in the earlier stage of training), which may lead to the wrong gradient.

We argue that the approximation of $\nabla_\theta \mathcal{L}(V, \theta)$ should: 1) be free from the overfitting problem; 2) be calculated with the reliable ground-truth label to ensure the correctness. In this paper, we propose a Mixup-based approximation method. Given the labeled training set $S$, we randomly sample $2K$ examples from it: $\tilde{S} = \{(x_i, y_i), i = 1 \ldots 2K\}$, followed by the backbone to extract features $h$ for each example: $h_{\tilde{V}} = \{h_i, i = 1 \ldots 2K\}$. Then, we apply Mixup to the feature and ground-truth label: $\overline{h_i} = \text{Mixup}(h_{2i}, h_{2i+1})$, $\overline{y_i} = \text{Mixup}(y_{2i}, y_{2i+1})$ to obtain the support set $\overline{S} = \{(\overline{h_i}, \overline{y_i}), i = 1 \ldots K\}$. Finally, the classification head will output the confidence distributions based on the features after Mixup and compute the loss. The gradient $\nabla_\theta \mathcal{L}(\overline{S}, \theta)$ is used as the approximation of $\nabla_\theta \mathcal{L}(V, \theta)$.

Mixup on labeled examples can provide accurate pseudo labels and alleviate the problem of overfitting. What's more, the feature-level Mixup ensures the input domain is unchanged, so that the backbone network can extract features correctly, making the gradient of support set closer to $\nabla_\theta \mathcal{L}(V, \theta)$.

## E  EXPERIMENTS DETAILS

As for VCC, we compare it with SimMatch(Zheng et al., 2022), FlexMatch(Zhang et al., 2021), Dash(Xu et al., 2021), MPL (Pham et al., 2021), FixMatch(Sohn et al., 2020), ReMixMatch(Berthelot et al., 2020), UDA(Xie et al., 2020), MixMatch(Berthelot et al., 2019), VAT(Miyato et al., 2019), MeanTeacher(Tarvainen & Valpola, 2017) and PL(Lee et al., 2013). As for core set selection experiments, we use FlexMatch as the SSL method and compare INFUSE with RETRIEVE(Killamsetty et al., 2021b), GradMatch(Killamsetty et al., 2021a), EL2N(Paul et al., 2021), Random(Killamsetty et al., 2021a), and Earlystop(Killamsetty et al., 2021a).

The model is trained under the most commonly used SSL setting (Sohn et al., 2020). The total number of iterations is $2^{20}$ (segmented into 1024 epochs) and batch-size of labeled/unlabeled data is 64/448. We use SGD to optimize the parameters. The learning rate is initially set as $\eta_0 = 0.03$ with a cosine learning rate decay schedule as $\eta = \eta_0 \cos\left(\frac{7\pi k}{16K}\right)$, where $k$ is the current iteration and $K$ is the total iterations.

As for VCC, the size of random noise $z$ is set as 16 for best performance. To reduce the computation overhead, the encoder $q_\phi$ and decoder $p_\theta$ are MLPs with 2 hidden layers (with dimensions 256 and 64). The hyperparameter $\lambda_{VCC}$ is set as 2.0.

In INFUSE, specifically, the core set is updated for every 40 epochs, and the total number of iterations is adjusted with the keep ratio $k$. Take $k = 10\%$ for example, the amount of examples in core set is $10\% \times |U|$ and the total steps is $10\% \times 2^{20}$.

The error rate on test set is used as the main metric to evaluate the effectiveness of our methods. To further study the reduction of calibration error, we introduce ECE/MCE (Guo et al., 2017) and ACE (Nixon et al., 2019). To calculate the calibration error, the examples in test set into buckets $B_1, B_2, \ldots, B_m$ based on confidence scores (for example, samples with confidence scores in the range $[0.90 \; 0.95)$ are assigned to the same bucket). The Expected Calibration Error (ECE) can be formulated as:

$$\text{ECE} = \sum_{i=1}^{m} \frac{|B_i|}{N} \left| \text{conf}(B_i) - \text{acc}(B_i) \right|, \tag{27}$$

where $\text{acc}(B_i)$ and $\text{conf}(B_i)$ represent the average accuracy and average confidence score of examples in bucket $B_i$, respectively.

Unlike ECE, Maximum Calibration Error (MCE) Guo et al. (2017) measures the model's calibration error in the worst-case scenario. It can be expressed as:

$$\text{MCE} = \max_i \left| \text{conf}(B_i) - \text{acc}(B_i) \right|, \tag{28}$$

The boundaries for dividing buckets in ECE and MCE are predefined confidence intervals. On the other hand, Adaptive Calibration Error (ACE) (Nixon et al., 2019) aims to ensure an equal number of samples in each bucket during grouping. By ensuring that each bucket contains $\lfloor \frac{n}{m} \rfloor$ samples, the resulting buckets can be denoted as $B'_1, B'_2, \ldots, B'_m$. The formula for ACE is as follows:

$$\text{ACE} = \sum_{i=1}^{m} \frac{|B'_i|}{N} \left| \text{conf}(B'_i) - \text{acc}(B'_i) \right|. \tag{29}$$

A model with lower ECE/MCE/ACE is expected to be better calibrated.

## F  SUPPLEMENTARY RESULTS

### F.1  COMPARE VCC WITH OTHER CALIBRATION METHODS

| Method | ER(%) | ECE | MCE | ACE |
|---|---|---|---|---|
| FlexMatch | 26.49 | 0.169 | 0.369 | 0.173 |
| FlexMatch + Ensemble-TS (Zhang et al., 2020) | 26.36 | 0.165 | 0.382 | 0.174 |
| FlexMatch + MMCE (Kumar et al., 2018) | 28.44 | 0.182 | 0.374 | 0.185 |
| VCC-FlexMatch (Ours) | **25.26** | **0.147** | **0.324** | **0.163** |

Table 8: The error rate of VCC and other calibration methods on CIFAR-100 dataset with 2500 labeled examples.

### F.2  THE ABLATION STUDIES OF THE THREE CONSISTENCY SCORES

| ensemble socre | temporal score | view score | ER(%) | ECE | MCE | ACE |
|---|---|---|---|---|---|---|
| ✓ | ✓ | ✗ | 25.45 | 0.148 | 0.328 | 0.167 |
| ✓ | ✗ | ✓ | 25.97 | 0.166 | 0.352 | 0.168 |
| ✗ | ✓ | ✓ | 25.65 | 0.153 | 0.337 | 0.169 |
| ✓ | ✓ | ✓ | **25.26** | **0.147** | **0.324** | **0.163** |

Table 9: The Error Rate (ER) and calibration errors of VCC when different consistency score is disabled while approximating the $\tilde{r}_u$. The results are based on the CIFAR-100 dataset with 2500 labeled examples.

## G  EXPERIMENTS SETTING OF IMBALANCE SSL

In this section, we explore a more difficult but realistic setting: imbalance semi-supervised learning. While the problem of SSL comes from the situation that it's difficult to make labels for all data, collecting balance data for each class could be more challenging since most ground-truth information is not accessible when making dataset. Using imbalance data to train model can make the bias generated during semi-supervised learning become more severe, thus hurting the performance.

We test the robustness of VCC on this problem to further testify its ability to reduce the bias and produce more accurate pseudo label. We construct a CIFAR-10-based long-tail distribution dataset in which the number of data points exponentially decreases from the largest to the smallest class, i.e., $N_k = N_1 \times \gamma^{-\frac{k-1}{L-1}}$, where $N_k$ stands for the number of labeled data of the k-th class, $\gamma = \frac{N_1}{N_L}$, and $L$ is the total class number. Let $\beta$ represent the ratio of labeled data to all data in the same class. We set $\beta = 20\%$, $N_1 = 1000$ and $\gamma = 100$ to build the long-tail dataset CIFAR-10-LT. We plug VCC into FixMatch and use the same setting as the main experiment to train the model on CIFAR-10-LT.

To better testify the effectiveness of VCC on this setting, we not only tarin FlexMatch, Fixmatch and SimMatch as comparisions, but also include other Imbalance Semi-supervised Learning methods that focus on exploiting the distribution bias information, such as DASO (Oh et al., 2022), DebiasPL (Wang et al., 2022), DARP (Kim et al., 2020) and Adsh (Guo & Li, 2022). We plug these Imbalance SSL methods into FixMatch for fairness and train them with their own default settings. The experiments have been given in Table 6.

