# OpenReview forum: "Boosting Semi-Supervised Learning via Variational Confidence Calibration and Unlabeled Sample Elimination"
_ICLR.cc/2024/Conference — Submitted to ICLR 2024_

### Official Review · Reviewer_iR75 · 2023-10-27

**Soundness:** 2 fair
**Presentation:** 3 good
**Contribution:** 2 fair
**Rating:** 5
**Confidence:** 5

**Summary:**

This study centers on pseudo labeling within the context of semi-supervised learning. To tackle the issue of inaccurate confidence scores and abundant unlabeled examples without data pruning, the author proposes two strategies of variational confidence calibration (VCC) and influence-function-based unlabeled sample elimination (INFUSE). Empirical assessments conducted on widely-adopted benchmark datasets demonstrate the efficacy of these proposed strategies. Notably, VCC yields a remarkable 3.16% reduction in error rates when compared to FixMatch.

**Strengths:**

1. The proposed strategies bolster pseudo labeling by addressing two key facets: computing dependable confidence scores and judiciously selecting a subset of the unlabeled dataset. These innovations result in a significant enhancement of generalization performance while also substantially reducing computational overhead in practical applications.
2. The writing is commendable, making the method easily comprehensible. The author offers ample experimental details, thereby facilitating the reproducibility of the study.

**Weaknesses:**

1. The novelty of the method appears somewhat constrained. Several components of the approach, such as Monte-Carlo Dropout, temporal consistency, exponential moving average, variational auto-encoder, and influence functions, are established techniques in the field.
2. The effectiveness of the Variational Autoencoder (VAE) implementation raises questions. VAE's main advantage lies in introducing randomness, and the efficacy of its calibration may require further substantiation. Additionally, the improvements achieved through VAE, as evidenced in Table 7, seem marginal at best.

**Questions:**

1. The author's proposal to generate ground-truth labels using a mixup-based method raises a valid concern about the dataset's stability during training. It's essential to verify whether the constructed labeled dataset remains invariant throughout the training process.
2. Table 9 in Appendix F.2 highlights the dominance of temporal scores in the experiments. It would be beneficial if the author could provide an explanation for this observation, shedding light on the reasons behind the temporal score's strong performance.
3. Suggestions for improvement: 1) Placing the table title above the table itself would enhance the document's readability. 2) Updating the template, especially for the page header, would contribute to a better presentation of the work.

---

> ### Author Response · Authors · 2023-11-20
>
> Thank you for your review and questions.
>
> **W1:** The novelty of the method appears somewhat constrained.
> **A1:** We admit that these techniques have been established in the literature. But the under-calibrated problem lacks enough attention in the SSL setting. We adopt three easy COSISTENCY measures to build calibration target under semi-supervised conditions. Furthermore, we make a step forward to leverage VAE to implicitly simulate the intractable factors. Although some components used in our method appear to be simple, our purpose is to make VCC a flexible and easy plug-in that can be combined with other SSL method. In addition, this work provides a new perspective by focusing on calibration in learning unreliable target. Finally, INFUSE leverages unreliable pseudo labels for selecting a small core set, which allows efficient training for time consuming SSL methods.
>
> **W2:** The effectiveness of the Variational Autoencoder (VAE) implementation raises questions.
> **A2:** We argue that the improvement obtained by VAE is remarkable. The accuracy improvement on accuracy is 1.23/% on CIFAR100 dataset for comparing VCC-FlexMatch with FlexMatch. This improvement is almost double when compared to using MLP simply. The overall results degenerate much if directly approaching the calibration target without VAE reconstructing (more than 2/% in some cases).
>
> **Q1:** The author's proposal to generate ground-truth labels using a mixup-based method raises a valid concern about the dataset's stability during training. It's essential to verify whether the constructed labeled dataset remains invariant throughout the training process.
> **A3:** The constructed dataset is invariant at the early stage of training, thus producing a stable selection. But at the later stage, we frequently switch the subset for preventing overfitting on unlabeled data. We will try to clarify this stability in detail in next edition.
>
> **Q2:** It would be beneficial if the author could provide an explanation for this observation, shedding light on the reasons behind the temporal score's strong performance.
> **A4:** Thank you for your valuable suggestion. All the three scores are useful for constructing a good calibration target, while the temporal consistency (TC) score helps to select unlabeled samples with more consistent prediction over time. As it is analyzed in previous work [1], selecting unlabeled data with less fluctuation prevents a rapid surge on the labeled sample loss, which explains the reason why temporal score can offer great benefits. Although our idea appears like previous work [1], there are two differences. First, a fixed moving window is used in our method, instead of EMA, to allow certain degree of forgetting and to permit easier confidence calibration. Second, the scores are combined to formulate the final calibration confidence target,  instead of selecting samples directly as used in [1].
>
> **Q3:** Suggestions for improvement.
> **A5:** Thank you for your valuable suggestions, and we will make these improvements in the next edition.
>
> [1] Tianyi Zhou, Shengjie Wang, and Jeff A. Bilmes. Time-Consistent Self-Supervision for SemiSupervised Learning. In International Conference on Machine Learning, Virtual, 2020.

---

> > ### Comment · Reviewer_iR75 · 2023-11-23
> >
> > Thank you for the detailed response, which addresses most concerns. However, I believe the work does not meet the acceptance criteria of ICLR in terms of novelty and contribution. I have chosen to keep the current score.

---

### Official Review · Reviewer_5hwM · 2023-10-29

**Soundness:** 3 good
**Presentation:** 2 fair
**Contribution:** 3 good
**Rating:** 6
**Confidence:** 4

**Summary:**

This manuscript proposes two methods, VCC and INFUSE, to improve semi-supervised learning by better utilizing unlabeled data. The effectiveness of these methods is demonstrated through experiments on multiple datasets. Overall, these methods offer promising solutions for improving the efficiency and accuracy of SSL.

**Strengths:**

Originality:
- The manuscript proposes two novel methods, VCC and INFUSE, to improve semi-supervised learning by better utilizing unlabeled data. These methods are designed to address the challenges of leveraging large-scale unlabeled data in SSL, and they offer promising solutions for improving the efficiency and accuracy of SSL.

Quality:
- The manuscript provides a thorough and well-organized presentation of the proposed methods, including detailed descriptions of the models, algorithms, and experiments. The experiments are conducted on multiple datasets and in various settings, and the results demonstrate the effectiveness of the proposed methods.

Significance:
- The proposed methods have the potential to significantly improve the efficiency and accuracy of SSL, which is an important and challenging problem in machine learning. The manuscript discusses the potential for extending the proposed methods to other SSL tasks, suggesting that they have broad applicability and potential impact in real-world scenarios.

**Weaknesses:**

1. The manuscript has some issues with the expression of details, making it difficult to follow. For example, the article does not provide an introduction to the first two loss terms in Eq. (11).
2. The latter part of the method involving INFUSE in the manuscript, and the earlier part on confidence calibration, seem to address two completely different problems, giving the paper a scattered feel and failing to highlight the main focus of the work. This leaves an impression of breadth over depth.
3. The author mentions that 'INFUSE uses the influence function from Koh & Liang (2017) to compute the importance of each unlabeled example', which implies that the solution to the second issue addressed in the manuscript merely references someone else's strategy. Both the problem itself and the method of solving it lack novelty.
4. The part on VIEW CONSISTENCY seems somewhat strained. Firstly, obtaining multiple views is difficult, and moreover, the EMA in the manuscript doesn't really have any connection with multiple views. EMA has already been showcased in the TEMPORAL CONSISTENCY section.
5. There is an issue in the reconstruction loss, where $\tilde{r}$ is treated as ground-truth; this itself is not accurate enough.

**Questions:**

1. I don't quite understand “we argue that the optimizing function in RETRIEVE only considers the loss on the labeled training set, which may lead to a deviation from the desired results (i.e. minimizing the loss on the validation set)”, could you please explain it in detail?
2. The author mentioned that " Although both consider the problem from the perspective of time, our temporal-consistency method is very dissimilar from the time-consistency method proposed by Zhou et al. (2020)." in Sec 3.2. Please give a detailed explanation and analysis.

---

> ### Author Response · Authors · 2023-11-20
>
> Thank you for your review and questions.
>
> **W1:** The manuscript has some issues with the expression of details, making it difficult to follow.
> **A1:** Thanks for your reminding, we are sorry for this and we promise to refine the details in the next version. As for the first two terms in Eq. 11, $\mathcal{L_{lab}}$ stands for the loss of labeled examples, which is a common cross entropy loss, and the $\mathcal{L_{unlab}}$ is the loss for unlabeled data, which is also a cross entropy loss, but implemented with the pseudo labels generated by SSL method.
>
> **W2:** The latter part of the method involving INFUSE in the manuscript, and the earlier part on confidence calibration, seem to address two completely different problems, giving the paper a scattered feel and failing to highlight the main focus of the work. This leaves an impression of breadth over depth.
> **A2:** The shared purpose of proposing VCC and INFUSE is to reduce the demanding need of accessing more data, but from two different perspectives. VCC can enhance the performance of SSL method when the number of labeled examples is small, such as only 4 labels per class for CIFAR10. This saves a lot of human labor for labeling. On the other hand, most pseudo-label-based SSL method is time consuming because of the multi-time inference and complex mechanism. INFUSE is proposed to largely reduce the time cost needed for training. Both of the two methods are trying to solving the dilemma that SSL method meets. Thank you for your reminding, and we will try to explain more about this motivation in next edition.
>
> **W3:** 'INFUSE uses the influence function from Koh & Liang (2017) to compute the importance of each unlabeled example'.
> **A3:** The original influence function is proposed under the fully-supervised setting with plenty of accessible labels. However, most of labels are unknown in SSL, so the original influence function is no longer suitable. The proposed INFUSE method leverages unlabeled data without ground-truth, so that the influence function is quite different from the original one. Previous SSL coreset selection method of RETRIEVE also makes some improvements based on the influence function. But a risk occurs to indicate a bias toward minority labeled subset because of its unreliable validation mechanism. Our INFUSE proposes a mix-up method to reduce this negative bias.
>
> **W4:** The part on VIEW CONSISTENCY seems somewhat strained.
> **A4:** We use the EMA scheme to prevent training multiple models from scratch when small subsets are provided to include only a few labeled examples in each. Note that accessible labeled data is extremely limited. The EMA scheme can be regarded as an easy way to obtain a host of different models with temporal variations. However, the differences of these obtained EMA models may be small, so the cross-feature trick is adopted to produce better multi-view models with great diversity. In summary, the purpose of VIEW CONSISTENCY is totally different from the TEMPORAL CONSISTENCY.
>
> **W5:** The target in reconstruction loss is not accurate enough.
> **A5:** We have already considered this inaccurate problem. Accordingly, we introduce VAE to employ hidden variables for enabling the reconstruction prediction, instead of directly approaching the reconstruction target. The experiment result justifies the effectiveness.
>
> **Q1:** “we argue that the optimizing function in RETRIEVE only considers the loss on the labeled training set, which may lead to a deviation from the desired results (i.e. minimizing the loss on the validation set)”?
> **A6:** RETRIEVE tries to select a subset of unlabeled data that achieves the lowest loss only on the labeled data, but the number of labeled data might be very small (e.g. 4 labeled examples per class in CIFAR100). In other words, the subset selected by RETRIEVE is not validated on the huge amount of unlabeled data. Therefore, the coreset obtained by RETRIEVE introduces large deviation on the overall dataset consisting of labeled and unlabeled examples. We will clarify this point in the next edition.
>
> **Q2:** " Although both consider the problem from the perspective of time, our temporal-consistency method is very dissimilar from the time-consistency method proposed by Zhou et al. (2020)."?
> **A7:** Our temporal consistency (TC) score is different from Zhou's in several ways. First, our temporal consistency score employs a fixed moving window instead of EMA used in Zhou's method. The fixed moving window permits to forget the past to some extent, which is beneficial to easier confidence calibration as the fundamental goal of VCC. Second, we combine the scores to formulate the final calibration confidence target for the model. Instead, Zhou's TC method directly selects examples with high TC scores into the training procedure.

---

### Official Review · Reviewer_o4va · 2023-10-30

**Soundness:** 4 excellent
**Presentation:** 4 excellent
**Contribution:** 3 good
**Rating:** 6
**Confidence:** 3

**Summary:**

This paper proposes a new semi-supervised learning technique which is based on "variational confidence calibration" (for calibrating the predictions on unlabeled examples) and "unlabeled sample elimination" (for pruning data with the goal to decrease the running time of the method). The main contributions of this paper are as follows:

(i) The authors propose the Variational Confidence Calibration (VCC) method, which aims to obtain well-calibrated scores for pseudo-label selection. The method is based on computing three different scores (ensemble consistency, temporal consistency view consistency), appropriately combining them, and feeding them to a (trainable) variational auto encoder  to get the final calibrated score. The resulting score can be used in combination with other standard/SOTA semi-supervised learning techniques.

(ii) The author propose the INFUSE method, which can dynamically prune unimportant unlabeled examples, in order to speed up  the convergence and reduce the computation costs in training.

(iii) Extensive experimental evaluation showing the competitiveness of the proposed method with respect to other SOTA methods.

**Strengths:**

— This is a well-written paper that proposes an interesting approach to semi-supervised learning.

— The use of the VAE in the computation of the calibrated scores is a novel and intriguing idea.

— Extensive experimental evaluation showing SOTA results in various datasets. In a few cases the performance gains (in terms of test-accuracy) are quite significant, e.g. in the CIFAR-100 dataset with 400 labeled examples VCC reduces the classification error rate of FixMatch from 46.47% to 43.31% (with improvement of 3.16%).

**Weaknesses:**

— The performance gains of using the method (in terms of test-accuracy) are typically somewhat mild, and often times less than 0.5%.

— The method seems to be a bit involved, especially given the typical overall benefit.

**Questions:**

Have the authors tried to apply their method in larger datasets like Imagenet? (I know that many of the SOTA semi-supervised learning approach suffer in the case of many classes/ large scale datasets this is why I am asking.)

---

> ### Author Response · Authors · 2023-11-20
>
> Thank you for your review. And hare are our responses.
>
> **W1:** The performance gains of using the method (in terms of test-accuracy) are typically somewhat mild, and often times less than 0.5%.
> **A1:** It's true that the gains are sometimes mild in terms of accuracy, but this phenomenon mostly happens in the cases when enough labels are provided. As we have mentioned in our paper, the notable advantage of our approach is that VCC can provide significant improvement compared with base algorithms when the number of available labels is small, such as the 3.16% improvement on FixMatch when only 4 labels of each class in CIFAR-100 is provided. The results in Table 2 can further justify this. Moreover, please see the new experimental results on Imagenet in response to your Question 1. Former methods like FlexMatch also suffers from decreasing improvement compared with base FixMatch when the labeled data increase, since the advantages of the 'novel design' decrease gradually. Besides, the test accuracy is not the only metric we considered in SSL. The calibration metrics like ECE are also introduced into this field, and the gain on these metrics should not be ignored when evaluating the effectiveness of VCC, since the under-calibrated problem is essential to SSL methods.
>
> **W2:** The method seems to be a bit involved, especially given the typical overall benefit.
> **A2:** VCC is a plug-in and can be combined with any pseudo-label-based SSL methods. Once the super-parameters are validated, the training strategy can be easily transferred to different methods. Due to this benefit, VCC and INFUSE reduce the amount of data and labels for any SSL methods. We do believe that the overall framework is valuable to the SSL community.
>
> **Q1:** Have the authors tried to apply their method in larger datasets like Imagenet?
> **A3:** We are sorry about this, but we only have one 3090 GPU to carry out all our experiments. We try to make a compensation by carrying out trials on ImageNet-32 based on FixMatch and FlexMatch with 100 labels per class (less than 8%), and the results are shown at below. We promise to provide more completed results in the next version.
>
> | method        | accuracy(%) |
> | --------      | ---------   |
> | FixMatch      | 32.06       |
> | VCC-FixMatch  | 36.53 (+4.47%)       |
> | FlexMatch     | 36.89       |
> | VCC-FlexMatch | 39.21 (+2.32%)       |

---

### Official Review · Reviewer_LFY9 · 2023-10-30

**Soundness:** 2 fair
**Presentation:** 3 good
**Contribution:** 2 fair
**Rating:** 5
**Confidence:** 4

**Summary:**

This paper studies semi-supervised learning (SSL). This paper points out two issues of existing SSL methods, including 1) the incorrect pseudo labels caused by calibration error, 2) the huge computation cost in training. To address the first issue, this paper proposes Variational Confidence Calibration (VCC), a variational method to obtain the calibrated confidence scores for pseudo-label selection. To address the second issue, this paper proposes the INfluence Function-based Unlabeled Sample Elimination (INFUSE) method, which uses the influence function to compute the importance of each unlabeled example. The two methods can be combined together to achieve high prediction accuracy with lower training costs. Experimental results demonstrate the effectiveness of the proposed methods.

**Strengths:**

- The writing is very clear.
- The proposed two methods are reasonable. There is an important advantage of VCC, i.e., it can be plugged into existing SSL methods to enhance their performance.
- Experimental results and ablation studies support the proposed methods.

**Weaknesses:**

- The proposed methods seem not novel enough, because they are only adapted from existing techniques, i.e., Variational Auto Encoder and Influence Function. It is intuitive that such a combination method can work well and thus I cannot see any important insights brought by the two methods.
- I do not think it is a good strategy to address two independent problems of SSL together, which may not increase the contributions of this paper. A good paper is supported by an important finding/contribution. Two independent minor contributions to address different issues may not form a single significant contribution. So I would suggest that the authors should focus on a major problem and try to solve this paper from another novel perspective to dig more deeply.
- In some tables, only a single result (without using mean$\pm$std) is provided. I suggest further providing standard deviations.

**Questions:**

Please check the above weaknesses.

---

> ### Author Response · Authors · 2023-11-20
>
> Thank you for your careful review.
> **W1:** The proposed methods seem not novel enough, because they are only adapted from existing techniques.
> **A1:** We believe that the introducing of two methods is not a plain adaption of existing methods. First, the under-calibrated problem is the improtant problem of SSL methods. While most existed calibration methods are designed for fully-supervised problem, we propose a reasonable strategy to generate calibration target under Semi-Supervised background. However, directly approaching this target is not reliable enough since there are many "nuisance" factors as we have mentioned in paper, and this is why we introduce VAE as a calibration tool to carry out a soft approaching to the sub-optimal target. Second, most existing data selection method is designed under fully-supervised settings and can not be directly used for SSL. To address this issue, this paper proposes the INFUSE method for selecting the core data set when unlabeled data can only have access to pseudo labels in SSL setup. We believe none of these methods have been thoroughly studied before for SSL problems.
>
> **W2:** Two independent minor contributions to address different issues may not form a single significant contribution.
> **A2:** Originally we started the study of two methods as two independent research lines at the very beginning, but later we find that there is a deep connection between them: they both further reduce the resources to train a good SSL method. To be more specific, the plug-in VCC can largely improve the accuracy of base algorithms with only a few labels, which saves a lot of effort for labeling. On the other hand, most existed SSL method requires multiple forward passes and complex mechanism to select data and to generate pseudo labels for training, which makes the training procedure remarkably time-consuming. The proposed INFUSE method can largely save the training time. We want to combine two resource-effective methods together to provide a unified framework. We will illustrate in details about the motivation above in our next edition.
>
> **W3:** In some tables, only a single result (without using mean std) is provided. I suggest further providing standard deviations.
> **A3:** Thank you for your suggestion. We are sorry for this, but we have tried our best to provide as more results as we could with limited GPU resources. We will provide further results as you suggest in next version.

---

> > ### Comment · Reviewer_LFY9 · 2023-11-22
> >
> > Thank you for providing the rebuttal. I feel that I am still not convinced. 1) It is indeed a combination of two existing techniques without new insights. 2) I do not agree with that "there is a deep connection between them because they both further reduce the resources to train a good SSL method". If this is true, many methods that have the same purpose can be said to be deeply connected. 3) I would expect the results, because standard deviations are really important.
> >
> > I would admit that the points I am concerned about are somewhat subjective. But personally, I feel that the novelty and contribution are not enough.

---

### Meta-Review · Area_Chair_fqfc · 2023-12-07

**Metareview:**

This paper addresses two issues in current SSL methods: 1) the incorrect pseudo labels caused by calibration error, and 2) the huge computation cost in training. The authors propose a method called Variational Confidence Calibration (VCC), a variational method to obtain the calibrated confidence scores for pseudo-label selection. To address the second issue, this paper proposes the INfluence Function-based Unlabeled Sample Elimination (INFUSE) method, which uses the influence function to compute the importance of each unlabeled example. The two methods can be combined together to achieve high prediction accuracy with lower training costs.

Most of the reviewers acknowledge the strengths in the writing and experimental aspects of the paper. However, they also point out several weaknesses: 1) The novelty of the method appears somewhat constrained. 2) The motivation of addressing two issues in one method is not convinced. 3) The performance gains of using the method are typically somewhat mild. Based on the overall reviews, I am inclined to reject this paper.

**Justification For Why Not Higher Score:**

1. The novelty of the method appears somewhat constrained. The proposed methods are adapted from existing techniques, i.e., Variational Auto Encoder and Influence Function. This combination of existing methods is intuitive and lacks of important insights brought by the two methods.

2. The motivation of addressing two issues in one method is not convinced. As mentioned by reviewer LFY9, a good paper is supported by an important finding/contribution. Two independent minor contributions to address different issues may not form a single significant contribution.

3. The performance gains of using the method (in terms of both accuracy and calibration error) are typically somewhat mild.

**Justification For Why Not Lower Score:**

N/A.

---

### Decision · Program_Chairs · 2024-01-16

Reject